# AGNOSTIC SHARPNESS-AWARE MINIMIZATION

## ABSTRACT

Sharpness-aware minimization (SAM) has been instrumental in improving deep neural network training by minimizing both the training loss and the sharpness of the loss landscape, leading the model into flatter minima that are associated with better generalization properties. In another aspect, Model-Agnostic Meta-Learning (MAML) is a framework designed to improve the adaptability of models. MAML optimizes a set of meta-models that are specifically tailored for quick adaptation to multiple tasks with minimal fine-tuning steps and can generalize well with limited data. In this work, we explore the connection between SAM and MAML in enhancing model generalization. We introduce Agnostic-SAM, a novel approach that combines the principles of both SAM and MAML. Agnostic-SAM adapts the core idea of SAM by optimizing the model toward wider local minima using training data, while concurrently maintaining low loss values on validation data. By doing so, it seeks flatter minima that are not only robust to small perturbations but also less vulnerable to data distributional shift problems. Our experimental results demonstrate that Agnostic-SAM significantly improves generalization over baselines across a range of datasets and under challenging conditions such as noisy labels or data limitation.

## 1 INTRODUCTION

Deep neural networks have become the preferred method for analyzing data, surpassing traditional machine learning models in complex tasks such as classification. These networks process input through numerous parameters and operations to predict classes. The learning process involves finding parameters within a model space that minimize errors or maximize performance for a given task. Typically, training data, denoted as $S$, is finite and drawn from an unknown true data distribution $\mathcal{D}$. Larger or more aligned training sets lead to more efficient models.

Despite their ability to learn complex patterns, deep learning models can also capture noise or random fluctuations in training data, leading to overfitting. This results in excellent performance on training data but poor predictions on new, unseen data, especially with domain shifts. Generalization, measured by comparing prediction errors on $S$ and $\mathcal{D}$, becomes crucial. Balancing a model's ability to fit training data with its risk of overfitting is a key challenge in machine learning.

Several studies have been done on this problem, both theoretically and practically. Statistical learning theory has proposed different complexity measures that are capable of controlling generalization errors (Vapnik, 1998; Bartlett & Mendelson, 2003; Mukherjee et al., 2002; Bousquet & Elisseeff, 2002; Poggio et al., 2004). In general, they develop a bound for general error on $\mathcal{D}$. Theory suggests that minimizing the intractable general error on $\mathcal{D}$ is equivalent to minimizing the empirical loss on $S$ with some constraints to the complexity of models and training size (Alquier et al., 2016b). An alternative strategy for mitigating generalization errors involves the utilization of an optimizer to learn optimal parameters for models with a specific local geometry. This approach enables models to locate wider local minima, known as flat minima, which makes them more robust against data shift between training and testing sets (Jiang et al., 2020; Petzka et al., 2021; Dziugaite & Roy, 2017).

The connection between a model's generalization and the width of minima has been investigated theoretically and empirically in many studies, notably (Hochreiter & Schmidhuber, 1994; Neyshabur et al., 2017; Dinh et al., 2017; Fort & Ganguli, 2019). A specific method within this paradigm is Sharpness-aware Minimisation (SAM) (Foret et al., 2021), which has emerged as an effective technique for enhancing the generalization ability of deep learning models. SAM seeks a perturbed

model within the vicinity of a current model that maximizes the loss over a training set. Eventually, SAM leads the model to the region where both the current model and its perturbation model have low loss values, which ensure flatness. The success of SAM and its variants (Kwon et al., 2021; Kim et al., 2022; Truong et al., 2023) has inspired further investigation into its formulation and behavior, as evidenced by recent works such as (Kaddour et al., 2022; Möllenhoff & Khan, 2022; Andriushchenko & Flammarion, 2022).

SAM significantly enhances robustness against shifts between training and testing datasets, thereby reducing overfitting and improving overall performance across different datasets and domains. This robust optimization approach aligns particularly well with the principles of Model-Agnostic Meta-Learning (MAML) (Finn et al., 2017). MAML aims to find a set of meta-model parameters that not only generalize well on current tasks but can also be quickly adapted to a wide range of new tasks. Furthermore, the agnostic perspective of MAML is particularly enticing for enhancing generalization ability because it endeavors to learn the optimal meta-model from meta-training sets capable of achieving minimal losses on independent meta-testing sets, thus harmonizing with the goal of generalization.

In this paper, inspired by MAML and leveraging SAM, we initially approach the problem of learning the best model over a training set from an agnostic viewpoint. Subsequently, we harness this perspective with sharpness-aware minimization to formulate an agnostic optimization problem. However, a naive solution akin to MAML does not suit our objectives. We propose a novel solution for this agnostic optimization problem, resulting in an approach called *AgnosticSAM*. In summary, our contributions to this work are as follows:

- We proposed a framework inspired by SAM and MAML works, called Agnostic-SAM to improve model flatness and robustness against noise. Agnostic-SAM updates a model to a region that minimizes the sharpness on the training set while also implicitly performing well on the validation set by using a combination of gradients on both training and validation sets.

- We demonstrate the effectiveness of Agnostic-SAM in improving generalization performance. Our initial examination focuses on image classification tasks, including training from scratch and transfer learning across a range of datasets, from small to large scale. We also extend this experiment under noisy label conditions with varying levels of noise. Additionally, we apply Agnostic-SAM in MAML settings to validate the effectiveness of our method in generalizing beyond the meta-training tasks and its adaptability across different domains. The consistent improvement in performance across experiments indicates that Agnostic-SAM not only enhances robustness against label noise and improves the model's generalization across diverse tasks, but also contributes to more stable and reliable predictions in different settings.

## 2 RELATED WORKS

**Sharpness-Aware Minimization.** The correlation between the wider minima and the generalization capacity has been extensively explored both theoretically and empirically in various studies (Jiang et al., 2020; Petzka et al., 2021; Dziugaite & Roy, 2017). Many works suggested that finding flat minimizers might help to reduce generrlization error and increase robustness to data distributional shift problems in various settings (Jiang et al., 2020; Petzka et al., 2021; Dziugaite & Roy, 2017). There are multiple works have explored the impact of different training parameters, including batch size, learning rate, gradient covariance, and dropout, on the flatness of discovered minima such as (Keskar et al., 2017; Jastrzebski et al., 2017; Wei et al., 2020).

Sharpness-aware minimization (SAM) (Foret et al., 2021) is a recent optimization technique designed to improve the generalization error of neural networks by considering the sharpness of the loss landscape during training. SAM minimizes the worst-case loss around the current model and effectively updates models towards flatter minima to achieve low training loss and maximize generalization performance on new and unseen data. SAM has been successfully applied to various tasks and domains, such as vision models (Chen et al., 2021), language models (Bahri et al., 2022), federated learning (Qu et al., 2022), Bayesian Neural Networks (Nguyen et al., 2023), domain generalization (Cha et al., 2021), multi-task learning (Phan et al., 2022) and meta-learning bi-level optimization (Abbas et al.,

2022). In Abbas et al. (2022), authors discussed SAM's effectiveness in enhancing meta-learning bi-level optimization, while SAM's superior convergence rates in federated learning compared to existing approaches in Qu et al. (2022) along with proposing a generalization bound for the global model. Additionally, multiple varieties of SAM have been proposed (Kwon et al., 2021), (Li et al., 2024), (Du et al., 2022) to tackle the different problems of the original method.

**Model-Agnostic Machine Learning.** Model-agnostic machine learning techniques have significant advances and offer flexible solutions applicable across various models and tasks. In which, MAML (Finn et al., 2017) stands out as the most compelling model-agnostic technique that formulates meta-learning as an optimization problem, enabling models to improve the model ability to quickly adapt to new tasks with minimal task-specific modifications or limited additional data. Subsequent research has largely focused on addressing the computational challenges of MAML (Chen et al., 2023; Wang et al., 2023) or proposing novel approaches that exploit the concept of model agnostic from MAML across a wide range of tasks, including non-stationary environments (Al-Shedivat et al., 2018), alternative optimization strategies (Rajeswaran et al., 2019), and uncertainty estimation for robust adaptation (Finn et al., 2018). Recently, Abbas et al. (2022) analyzed the loss-landscape of MAML models and proposed the integration of SAM in training to improve the generalization performance of a meta-model.

## 3 PROPOSED FRAMEWORK

### 3.1 NOTIONS

We start by introducing the notions used throughout our paper. We denote $\mathcal{D}$ as the data/label distribution to generate pairs of data/label $(x, y)$. Given a model with the model parameter $\theta$, we denote the per sample loss induced by $(x, y)$ as $\ell(x, y; \theta)$. Let $S$ be the training set drawn from the distribution $\mathcal{D}$. We denote the empirical and general losses as $\mathcal{L}_S(\theta) = \mathbb{E}_S[\ell(x, y; \theta)]$ and $\mathcal{L}_\mathcal{D}(\theta) = \mathbb{E}_\mathcal{D}[\ell(x, y; \theta)]$ respectively. We define $\mathcal{L}_\mathcal{D}(\theta \mid S)$ as an *upper bound defined over $S$* of the general loss $\mathcal{L}_\mathcal{D}(\theta)$. Note that inspired by SAM (Foret et al., 2021), we use the sharpness over $S$ to define $\mathcal{L}_\mathcal{D}(\theta \mid S)$. Finally, we use $|A|$ to denote the cardinality of a set $A$.

### 3.2 PROBLEM FORMULATION

Given a training set $S^t$ whose examples are sampled from $\mathcal{D}$ (i.e., $S^t \sim \mathcal{D}^{N_t}$ with $N_t = |S^t|$), we use $\mathcal{L}_\mathcal{D}(\cdot \mid S^t)$ to train models. Among the models that minimize this loss, we select the one that minimizes the general loss as follows:

$$\min_{\theta^*} \mathcal{L}_\mathcal{D}(\theta^*) \text{ s.t. } \theta^* \in \mathcal{A}_\mathcal{D}(S^t) = \text{argmin}_\theta \mathcal{L}_\mathcal{D}(\theta \mid S^t). \tag{1}$$

The reason for the formulation in (1) is that although $\mathcal{L}_\mathcal{D}(\theta \mid S^t)$ is an upper bound of the general loss $\mathcal{L}_\mathcal{D}(\theta)$, there always exists a gap between them. Therefore, the additional outer minimization helps to refine the solutions. We now denote $S^v$ (i.e., $S^v \sim \mathcal{D}^{N_v}$ with $N_v = |S^v|$) as a valid set sampled from $\mathcal{D}$. With respect to this valid set, we have the following theorem.

**Theorem 1.** *Denote $\mathcal{L}_\mathcal{D}(\theta \mid S) := \max_{\theta':\|\theta'-\theta\|_2 \leq \rho} \mathcal{L}_S(\theta')$. Under some mild condition similar to SAM (Foret et al., 2021), with a probability greater than $1 - \delta$ (i.e., $\delta \in [0, 1]$) over the choice of $S^v \sim \mathcal{D}^{N_v}$, we then have for any optimal models $\theta^* \in \mathcal{A}_\mathcal{D}(S^t)$:*

$$\mathcal{L}_\mathcal{D}(\theta^*) \leq \mathcal{L}_\mathcal{D}(\theta^* \mid S^v) + \frac{4L}{\sqrt{N_v}} \left[ k \log \left( 1 + \frac{\|\theta^*\|^2}{\rho} \left( 1 + \sqrt{\log N_v / k} \right) \right) + 2\sqrt{\log \frac{N_v + k}{\delta}} + O(1) \right], \tag{2}$$

*where $L$ is the upper-bound of the loss function (i.e., $\ell(x, y; \theta) \leq L, \forall x, y, \theta$), $k$ is the model size, and $\rho > 0$ is the perturbation radius.*

Our theorem 1 (proof can be found in Appendix A.1) can be viewed as an extension of Theorem 1 in Foret et al. (2021), where we apply the Bayes-PAC theorem from Alquier et al. (2016a) to prove an upper bound for the general loss of any bounded loss, instead of the 0-1 loss in Foret et al. (2021).

We can generalize this proof for $S^t$ to explain why we use $\mathcal{L}_{\mathcal{D}}\left(\theta \mid S^t\right) := \max_{\theta':\|\theta'-\theta\|_2 \leq \rho} \mathcal{L}_{S^t}\left(\theta'\right)$ as an objective to minimize, as in (1). Based on Theorem 1, we can rewrite the objectives in (1) as:

$$\min_{\theta^*} \mathcal{L}_{\mathcal{D}}\left(\theta^* \mid S^v\right) \text{ s.t. } \theta^* \in \mathcal{A}_{\mathcal{D}}\left(S^t\right) = \operatorname{argmin}_{\theta} \mathcal{L}_{\mathcal{D}}\left(\theta \mid S^t\right), \tag{3}$$

where $\mathcal{L}_{\mathcal{D}}\left(\theta \mid S\right) := \max_{\theta':\|\theta'-\theta\|_2 \leq \rho} \mathcal{L}_S\left(\theta'\right)$.

### 3.3 OUR SOLUTION

Our motivation here is to primarily optimize the loss over the training set $S^t$, while using $S^v$ to further enhance the generalization ability. Our agnostic formulation has the same form as MAML (Finn et al., 2017), developed for meta-learning. Inspired by MAML, a naive approach would be to consider $\theta^* = \theta^*(\theta)$ and then minimize $\mathcal{L}_{\mathcal{D}}\left(\theta^*\left(\theta\right) \mid S^v\right)$ with respect to $\theta$. However, this naive approach does not align with our objective, as it mainly focuses on optimizing the loss $\mathcal{L}_{\mathcal{D}}\left(\theta^*\left(\theta\right) \mid S^v\right)$ over the validation set $S^v$.

We interpret the bi-level optimization problem in (3) as follows: at each iteration, our primary objective is to optimize $\mathcal{L}_{\mathcal{D}}\left(\theta \mid S^t\right)$, primarily based on its gradients, in such a way that future models are able to implicitly perform well on $S^v$. To achieve this, similar to SAM (Foret et al., 2021), we approximate $\mathcal{L}_{\mathcal{D}}\left(\theta \mid S^t\right) = \max_{\|\theta'-\theta\| \leq \rho} \mathcal{L}_{S^t}\left(\theta'\right) \approx \mathcal{L}_{S^t}\left(\theta + \eta_1 \nabla \mathcal{L}_{S^t}\left(\theta\right)\right)$ for a sufficient small learning rate $\eta_1 > 0$ (i.e., $\eta_1 \|\nabla \mathcal{L}_{S^t}\left(\theta\right)\| \leq \rho$) and $\mathcal{L}_{\mathcal{D}}\left(\theta \mid S^v\right) = \max_{\|\theta'-\theta\| \leq \rho} \mathcal{L}_{S^v}\left(\theta'\right) \approx \mathcal{L}_{S^v}\left(\theta + \eta_2 \nabla \mathcal{L}_{S^v}\left(\theta\right)\right)$ for a sufficient small learning rate $\eta_2 > 0$ (i.e., $\eta_2 \|\nabla \mathcal{L}_{S^v}\left(\theta\right)\| \leq \rho$).

At each iteration, while primarily using the gradients of $\mathcal{L}_{\mathcal{D}}\left(\theta \mid S^t\right)$ for optimization, we also utilize the gradient of $\mathcal{L}_{\mathcal{D}}\left(\theta \mid S^v\right)$ in an auxiliary manner to ensure congruent behavior between these two gradients. Specifically, at the $l$-th iteration, we update as follows:

$$\tilde{\theta}_l^v = \theta_l + \eta_2 \nabla_\theta \mathcal{L}_{B^v}\left(\theta_l\right), \tag{4}$$

$$\tilde{\theta}_l^t = \theta_l + \eta_1 \nabla_\theta \mathcal{L}_{B^t}\left(\theta_l\right) - \eta_2 \nabla_\theta \mathcal{L}_{B^v}\left(\tilde{\theta}_l^v\right), \tag{5}$$

$$\theta_{l+1} = \theta_l - \eta \nabla_\theta \mathcal{L}_{B^t}\left(\tilde{\theta}_l^t\right), \tag{6}$$

where $\eta_1 > 0, \eta_2 > 0$, and $\eta > 0$ are the learning rates, while $\mathcal{L}_{B^t}\left(\theta_l\right)$ and $\mathcal{L}_{B^v}\left(\theta_l\right)$ represent the empirical losses over the mini-batches $B^t \sim S^t$ and $B^v \sim S^v$ respectively.

According to the update in (6) (i.e., $\theta_{l+1} = \theta_l - \eta \nabla_\theta \mathcal{L}_{B^t}\left(\tilde{\theta}_l^t\right)$), $\theta_{l+1}$ is updated to minimize $\mathcal{L}_{B^t}\left(\tilde{\theta}_l^t\right)$. We now do first-order Taylor expansion for $\mathcal{L}_{B^t}\left(\tilde{\theta}_l^t\right)$ as

$$\mathcal{L}_{B^t}\left(\tilde{\theta}_l^t\right) = \mathcal{L}_{B_t}\left(\theta_l\right) + \eta_1 \|\nabla_\theta \mathcal{L}_{B^t}\left(\theta_l\right)\|_2^2 - \eta_2 \nabla_\theta \mathcal{L}_{B^t}\left(\theta_l\right) \cdot \nabla_\theta \mathcal{L}_{B^v}\left(\tilde{\theta}_l^v\right), \tag{7}$$

where $\cdot$ specifies the dot product.

From (7), we reach the conclusion that the update in (6) (i.e., $\theta_{l+1} = \theta_l - \eta \nabla_\theta \mathcal{L}_{B^t}\left(\tilde{\theta}_l^t\right)$) aims to *minimize* simultaneously (i) $\mathcal{L}_{B_t}\left(\theta_l\right)$, (ii) $\|\nabla_\theta \mathcal{L}_{B^t}\left(\theta_l\right)\|_2^2$, and *maximize* (iii) $\nabla_\theta \mathcal{L}_{B^t}\left(\theta_l\right) \cdot \nabla_\theta \mathcal{L}_{B^v}\left(\tilde{\theta}_l^v\right)$. While the effects in (i) and (ii) are similar to SAM (Foret et al., 2021), maximizing $\nabla_\theta \mathcal{L}_{B^t}\left(\theta_l\right) \cdot \nabla_\theta \mathcal{L}_{B^v}\left(\tilde{\theta}_l^v\right)$) encourages two gradients of the losses over $B^t$ and $B^v$ to become more congruent.

**Theorem 2.** *For sufficiently small learning rates* $\eta_1 \leq \dfrac{\left|\nabla_\theta \mathcal{L}_{B_t}\left(\theta_l\right) \cdot \nabla_\theta \mathcal{L}_{B^v}\left(\tilde{\theta}_l^v\right)\right|}{12\left|\nabla_\theta \mathcal{L}_{B^v}\left(\tilde{\theta}_l^v\right)^T H_{B^t}\left(\theta_l\right) \nabla_\theta \mathcal{L}_{B^t}\left(\theta_l\right)\right|}$ *and* $\eta_2 \leq$

$\min\left\{\dfrac{\left|\nabla_\theta \mathcal{L}_{B_t}\left(\theta_l\right) \cdot \nabla_\theta \mathcal{L}_{B^v}\left(\tilde{\theta}_l^v\right)\right|}{6\left|\nabla_\theta \mathcal{L}_{B^v}\left(\tilde{\theta}_l^v\right)^T H_{B^t}\left(\theta_l\right) \nabla_\theta \mathcal{L}_{B^v}\left(\tilde{\theta}_l^v\right)\right|}, \dfrac{\left|\nabla_\theta \mathcal{L}_{B_t}\left(\theta_l\right) \cdot \nabla_\theta \mathcal{L}_{B^v}\left(\tilde{\theta}_l^v\right)\right|}{6\left|\nabla_\theta \mathcal{L}_{B^v}\left(\tilde{\theta}_l^v\right)^T H_{B^v}\left(\tilde{\theta}_l^v\right) \nabla_\theta \mathcal{L}_{B^t}\left(\theta_l\right)\right|}\right\}$, *we have*

$$\nabla_\theta \mathcal{L}_{B^t}\left(\tilde{\theta}_l^t\right) \cdot \nabla_\theta \mathcal{L}_{B^v}\left(\tilde{\theta}_l^v\right) \geq \begin{cases} \frac{1}{2}\nabla_\theta \mathcal{L}_{B^t}\left(\theta_l\right) \cdot \nabla_\theta \mathcal{L}_{B^v}\left(\tilde{\theta}_l^v\right) & if \nabla_\theta \mathcal{L}_{B^t}\left(\theta_l\right) \cdot \nabla_\theta \mathcal{L}_{B^v}\left(\tilde{\theta}_l^v\right) \geq 0 \\ \frac{3}{2}\nabla_\theta \mathcal{L}_{B^t}\left(\theta_l\right) \cdot \nabla_\theta \mathcal{L}_{B^v}\left(\tilde{\theta}_l^v\right) & otherwise \end{cases}$$

$$\tag{8}$$

Theorem 2 (proof can be found in Appendix A.1) indicates that two gradients $\nabla_\theta \mathcal{L}_{B^t} \left( \tilde{\theta}_l^t \right)$ and $\nabla_\theta \mathcal{L}_{B^v} \left( \tilde{\theta}_l^v \right)$ are encouraged to be more congruent since our update aims to maximize its lower bound $c \times \nabla_\theta \mathcal{L}_{B^t} (\theta_l) \cdot \nabla_\theta \mathcal{L}_{B^v} \left( \tilde{\theta}_l^v \right)$ (i.e., $c = 0.5$ or $c = 1.5$). Notice that the negative gradient $-\eta \nabla_\theta \mathcal{L}_{B^t} \left( \tilde{\theta}_l^t \right)$ is used to update $\theta_l$ to $\theta_{l+1}$, hence this update can have an implicit impact on minimizing $\mathcal{L}_{\mathcal{D}} (\theta \mid S^v)$ since the negative gradient $-\nabla_\theta \mathcal{L}_{B^v} \left( \tilde{\theta}_l^v \right)$ targets to minimize $\mathcal{L}_{\mathcal{D}} (\theta \mid S^v) = \max_{\|\theta' - \theta\| \leq \rho} \mathcal{L}_{S^v} (\theta') \approx \mathcal{L}_{S^v} (\theta + \eta_2 \nabla \mathcal{L}_{S^v} (\theta))$.

**Practical Algorithm.** Inspired by SAM Foret et al. (2021), we set $\eta_1 = \rho_1 \frac{\nabla_\theta \mathcal{L}_{B^t} (\theta_l)}{\|\nabla_\theta \mathcal{L}_{B^t} (\theta_l)\|_2}$ and $\eta_2 = \rho_2 \frac{\nabla_\theta \mathcal{L}_{B^v} (\theta_l)}{\|\nabla_\theta \mathcal{L}_{B^v} (\theta_l)\|_2}$, where $\rho_1 > 0$ and $\rho_2 > 0$ are perturbation radius. Furthermore, instead of splitting the training set $S$ into two fixed subsets, $S^t$ and $S^v$, which reduces the number of training samples, we set $S^t = S^v = S$, allowing the entire training set to be used for updating the model. This approach is especially beneficial for training on small datasets. Optionally, we apply momentum with a factor $\beta$ to approximate the gradient of the full validation set using gradients from mini-batches. The effectiveness of this term will be discussed in section 5.

The pseudo-code of Agnostic-SAM is summarized in Algorithm 1.

---

**Algorithm 1** Pseudo-code of Agnostic-SAM

---

**Input:** $\rho_1, \rho_2, \eta, \beta$, the number of iterations $L_{\text{iter}}$, and the training set $S$. Initialize gradient on the validation set $g_v \leftarrow 0$
**Output:** the optimal model $\theta_L$.
**for** $l = 1$ to $L_{\text{iter}}$ **do**
    Sample mini-batch $B^t \sim S^t$, $B^v \sim S^v$.
    Compute $\tilde{\theta}_l^v = \theta_l + \rho_2 \frac{\nabla_\theta \mathcal{L}_{B^v} (\theta_l)}{\|\nabla_\theta \mathcal{L}_{B^v} (\theta_l)\|_2}$
    $g_v \leftarrow \beta g_v + (1 - \beta) \nabla_\theta \mathcal{L}_{B^v} \left( \tilde{\theta}_l^v \right)$
    Compute $\tilde{\theta}_l^t \leftarrow \theta_l + \rho_1 \frac{\nabla_\theta \mathcal{L}_{B^t} (\theta_l)}{\|\nabla_\theta \mathcal{L}_{B^t} (\theta_l)\|_2} - \rho_2 \frac{g_v}{\|g_v\|_2}$.
    Compute $\theta_{l+1} \leftarrow \theta_l - \eta \nabla_\theta \mathcal{L}_{B^t} \left( \tilde{\theta}_l^t \right)$.
**end for**

---

## 4 EXPERIMENTS

In this section, we present the results of various experiments to evaluate the effectiveness of our Agnostic-SAM, including training from scratch, transfer learning on different dataset sizes, learning with noisy labels, and MAML setting. For all experiments of Agnostic-SAM, we consistently use a fixed value of momentum factor $\beta = 0.9$ and mini-batch size of validation set $4|B^v| = |B^t|$. The effectiveness of these hyper-parameters on performance and training complexity will be discussed in Section 5.

### 4.1 IMAGE CLASSIFICATION FROM SCRATCH

We first conduct experiments on ImageNet, Food101, and CIFAR datasets with standard image classification settings trained from scratch. The performance is compared with baseline models trained with the SGD, SAM, ASAM, and the integration of ASAM and Agnostic-SAM.

**ImageNet dataset** We use ResNet18 and ResNet34 models for experiments on the ImageNet dataset, with an input size of $224 \times 224$. For all experiments of Agnostic-SAM and its variations, we consistently set $\rho_1 = 2\rho_2 = 2\rho$, where $\rho$ represents the perturbation radius for the respective SAM method. Specifically, in this experiment, we set $\rho = 0.1$, $\rho_1 = 0.2$, and $\rho_2 = 0.1$. The models are trained for 200 epochs with basic data augmentations (random cropping, horizontal flipping, and

normalization). We use an initial learning rate of 0.1, a batch size of 2048 for the training set, and 512 for the validation set, following a cosine learning schedule across all experiments in this paper. We extend this experiment to the mid-sized Food101 dataset using the same settings, except for a batch size of 128 for the training set and 32 for the validation set. Performance results are detailed in Table 1.

Table 1: Classification accuracy on the ImageNet and Food101 datasets. All models are trained from scratch with 200 epochs.

| Dataset | Method | Resnet18 | | Resnet34 | |
|---|---|---|---|---|---|
| | | Top-1 | Top-5 | Top-1 | Top-5 |
| ImageNet | SAM | 62.46 | 84.19 | 63.73 | 84.95 |
| | Agnostic-SAM | **63.64** | **85.22** | **65.89** | **86.84** |
| Food101 | SAM | 73.15 | 89.85 | 73.87 | 90.84 |
| | Agnostic-SAM | **73.45** | **90.35** | **74.47** | **91.27** |

**CIFAR dataset**  We used three architectures: WideResnet28x10, Pyramid101, and Densenet121 with an input size of $32 \times 32$ for CIFAR datasets. To replicate the baseline experiments, we followed the hyperparameters provided in the original papers. Specifically, for CIFAR-100, we set $\rho = 0.1$, $\rho_1 = 0.2$, and $\rho_2 = 0.1$, and for CIFAR-10, we used $\rho = 0.05$, $\rho_1 = 0.1$, and $\rho_2 = 0.05$. The same procedure and settings were applied to ASAM and Agnostic-ASAM, with the perturbation radius $\rho$ for ASAM being 10 times larger than that of the SAM method. Other training configurations are consistent with those used in the ImageNet experiments, except for data augmentations (horizontal flipping, four-pixel padding, and random cropping). Results are reported in Tables 2, while the SGD results are referenced from Foret et al. (2021).

Our proposed method consistently outperforms the baselines across various settings. On both ImageNet and Food101, it significantly surpasses the baselines, with a notable improvement in both Top-1 and Top-5 accuracy. For CIFAR-10, performance is close to the saturation point, making further improvements challenging. Nevertheless, Agnostic-SAM achieves slight enhancements across all cases. On CIFAR-100, where models are more prone to overfitting compared to CIFAR-10, Agnostic-SAM still delivers competitive results.

Table 2: Classification accuracy on the CIFAR datasets. All models are trained from scratch three times with different random seeds and we report the mean and standard deviation of accuracies.

| Dataset | Method | WideResnet28x10 | Pyramid101 | Densenet121 |
|---|---|---|---|---|
| CIFAR-100 | SGD | $81.20 \pm 0.200$ | $80.30 \pm 0.300$ | - |
| | SAM | $83.00 \pm 0.035$ | $81.99 \pm 0.636$ | $68.72 \pm 0.409$ |
| | **Agnostic-SAM** | $\mathbf{83.49 \pm 0.049}$ | $\mathbf{82.38 \pm 0.282}$ | $\mathbf{69.10 \pm 0.311}$ |
| | ASAM | $83.16 \pm 0.296$ | $82.02 \pm 0.134$ | $69.62 \pm 0.120$ |
| | Agnostic-ASAM | $\mathbf{83.68 \pm 0.042}$ | $\mathbf{82.29 \pm 0.183}$ | $\mathbf{69.79 \pm 0.339}$ |
| CIFAR-10 | SGD | $96.50 \pm 0.100$ | $96.00 \pm 0.100$ | - |
| | SAM | $96.87 \pm 0.027$ | $96.17 \pm 0.174$ | $91.28 \pm 0.241$ |
| | **Agnostic-SAM** | $\mathbf{96.88 \pm 0.007}$ | $\mathbf{96.47 \pm 0.219}$ | $\mathbf{91.31 \pm 0.707}$ |
| | ASAM | $96.91 \pm 0.063$ | $96.45 \pm 0.042$ | $\mathbf{92.04 \pm 0.240}$ |
| | Agnostic-ASAM | $\mathbf{97.15 \pm 0.063}$ | $\mathbf{96.73 \pm 0.261}$ | $92.02 \pm 0.000$ |

## 4.2 TRANSFER LEARNING

In this subsection, we further evaluate Agnostic-SAM in the transfer learning setting using the ImageNet pre-trained models to fine-tune both small-size, mid-size, and large-size datasets. All initialized weights are available on the Pytorch library.

Table 3: Transfer learning on ImageNet with Resnet models.

| Model | Top-1 Acc | | Top-5 Acc | |
|---|---|---|---|---|
| | SAM | Agnostic-SAM | SAM | Agnostic-SAM |
| Resnet18 | 70.52 | **70.88** | 89.60 | **89.94** |
| Resnet34 | 73.06 | **73.84** | 91.29 | **91.81** |
| Resnet50 | 75.17 | **75.91** | 92.58 | **92.83** |

First, we conduct experiments on ImageNet by using three models from the Resnet family. These base models are both pre-trained on ImageNet by SGD and then fine-tuned for 50 epochs by SAM or Agnostic-SAM with a learning rate of 0.01. We $\rho = 0.05$ for SAM, and $\rho_1 = 2\rho_2 = 0.1$ for Agnostic-SAM and basic augmentation techniques, which are the same as training from the scratch setting. Results reported in Table 3 show that our methods outperform baselines with a significant gap in both top-1 and top-5 accuracies.

Next, we examine this setting on small and mid-sized datasets on three models of the EfficientNet family. We fine-tune with a learning rate of 0.05 in 50 epochs and use $\rho = 0.1$ for all experiments of SAM (as accuracies tend to decrease when reducing $\rho$), $\rho_1 = 2\rho_2 = 0.1$ for all experiments of Agnostic-SAM. In Table 4, Agnostic-SAM achieves a noticeable improvement compared to most of the baselines on all small-size, mid-size, and large-size datasets, demonstrating its robustness and stability across various experiment settings.

## 4.3 TRAIN WITH NOISY LABEL

In addition to mitigating data shifts between training and testing datasets, we evaluate the robustness of Agnostic-SAM against noisy labels on standard training procedure. Specifically, we adopt a classical noisy-label setting for CIFAR-10 and CIFAR-100, in which a portion of the training set's labels are symmetrically flipped with noise fractions {0.2, 0.4, 0.6, 0.8}, while the testing set's labels remain unchanged.

Table 4: Transfer learning accuracy of small and medium datasets. All models are fine-tuned from pre-trained weights on ImageNet.

| Dataset | Top-1 Acc | | | Top-5 Acc | | |
|---|---|---|---|---|---|---|
| | SGD | SAM | Agnostic-SAM | SGD | SAM | Agnostic-SAM |
| **EfficientNet-B2** | | | | | | |
| Stanford Cars | $89.14 \pm 0.11$ | $89.68 \pm 0.17$ | $\mathbf{90.34 \pm 0.07}$ | $97.60 \pm 0.20$ | $98.04 \pm 0.07$ | $\mathbf{98.24 \pm 0.09}$ |
| FGVC-Aircraft | $85.83 \pm 0.23$ | $86.25 \pm 0.36$ | $\mathbf{87.27 \pm 0.27}$ | $95.72 \pm 0.02$ | $95.87 \pm 0.06$ | $\mathbf{96.05 \pm 0.03}$ |
| Oxford IIIT Pets | $92.17 \pm 0.19$ | $92.34 \pm 0.11$ | $\mathbf{92.58 \pm 0.17}$ | $99.23 \pm 0.02$ | $99.35 \pm 0.02$ | $\mathbf{99.35 \pm 0.07}$ |
| Flower102 | $95.06 \pm 0.01$ | $95.22 \pm 0.14$ | $\mathbf{95.56 \pm 0.10}$ | $99.08 \pm 0.18$ | $99.11 \pm 0.19$ | $\mathbf{99.27 \pm 0.02}$ |
| Food101 | $83.50 \pm 0.01$ | $85.12 \pm 0.07$ | $\mathbf{85.51 \pm 0.02}$ | $96.10 \pm 0.32$ | $96.83 \pm 0.08$ | $\mathbf{97.14 \pm 0.00}$ |
| Country211 | $11.94 \pm 0.14$ | $12.48 \pm 0.03$ | $\mathbf{13.28 \pm 0.00}$ | $23.70 \pm 0.13$ | $25.49 \pm 0.07$ | $\mathbf{26.95 \pm 0.16}$ |
| **EfficientNet-B3** | | | | | | |
| Stanford Cars | $89.01 \pm 0.19$ | $89.40 \pm 0.09$ | $\mathbf{90.09 \pm 0.14}$ | $97.73 \pm 0.21$ | $98.03 \pm 0.07$ | $\mathbf{98.13 \pm 0.01}$ |
| FGVC-Aircraft | $84.88 \pm 0.08$ | $85.19 \pm 0.11$ | $\mathbf{85.99 \pm 0.25}$ | $95.53 \pm 0.12$ | $95.67 \pm 0.00$ | $\mathbf{96.08 \pm 0.10}$ |
| Oxford IIIT Pets | $92.68 \pm 0.25$ | $92.58 \pm 0.02$ | $\mathbf{92.75 \pm 0.19}$ | $99.00 \pm 0.01$ | $99.19 \pm 0.05$ | $\mathbf{99.20 \pm 0.11}$ |
| Flower102 | $94.59 \pm 0.10$ | $94.73 \pm 0.14$ | $\mathbf{95.16 \pm 0.26}$ | $98.95 \pm 0.08$ | $99.12 \pm 0.16$ | $\mathbf{99.18 \pm 0.07}$ |
| Food101 | $83.75 \pm 0.12$ | $85.79 \pm 0.13$ | $\mathbf{86.17 \pm 0.13}$ | $96.22 \pm 0.02$ | $97.12 \pm 0.00$ | $\mathbf{97.38 \pm 0.00}$ |
| Country211 | $12.96 \pm 0.01$ | $13.38 \pm 0.09$ | $\mathbf{13.63 \pm 0.05}$ | $26.11 \pm 0.56$ | $25.78 \pm 0.08$ | $\mathbf{26.71 \pm 0.26}$ |
| **EfficientNet-B4** | | | | | | |
| Stanford Cars | $84.72 \pm 0.04$ | $85.08 \pm 0.16$ | $\mathbf{85.79 \pm 0.32}$ | $96.41 \pm 0.07$ | $96.45 \pm 0.01$ | $\mathbf{96.77 \pm 0.00}$ |
| FGVC-Aircraft | $79.95 \pm 0.61$ | $79.96 \pm 0.04$ | $\mathbf{80.80 \pm 0.51}$ | $94.87 \pm 0.08$ | $94.65 \pm 0.08$ | $\mathbf{94.95 \pm 0.01}$ |
| Oxford IIIT Pets | $91.89 \pm 0.13$ | $92.02 \pm 0.23$ | $\mathbf{92.02 \pm 0.00}$ | $99.28 \pm 0.10$ | $99.43 \pm 0.07$ | $\mathbf{99.44 \pm 0.02}$ |
| Flower102 | $92.73 \pm 0.04$ | $93.02 \pm 0.14$ | $\mathbf{93.03 \pm 0.16}$ | $98.49 \pm 0.07$ | $\mathbf{98.68 \pm 0.02}$ | $98.59 \pm 0.05$ |
| Food101 | $84.55 \pm 0.14$ | $86.13 \pm 0.06$ | $\mathbf{86.15 \pm 0.44}$ | $96.31 \pm 0.03$ | $97.07 \pm 0.01$ | $\mathbf{97.22 \pm 0.02}$ |
| Country211 | $14.63 \pm 0.09$ | $14.80 \pm 0.13$ | $\mathbf{15.10 \pm 0.16}$ | $27.60 \pm 0.00$ | $29.09 \pm 1.77$ | $\mathbf{28.38 \pm 0.14}$ |

Table 5: Results under label noise on CIFAR dataset with ResNet32. Each experiment is conducted three times using different random seeds, and we report the average and standard deviation of the results.

| Method | Noise rate (%) | | | |
|---|---|---|---|---|
| | 0.2 | 0.4 | 0.6 | 0.8 |
| **Dataset CIFAR-100** | | | | |
| SGD | $66.22 \pm 0.355$ | $59.26 \pm 0.045$ | $46.77 \pm 0.020$ | $26.49 \pm 0.640$ |
| SAM | $66.16 \pm 0.721$ | $59.95 \pm 0.622$ | $50.81 \pm 0.353$ | $24.26 \pm 1.209$ |
| FSAM | $65.73 \pm 0.219$ | $58.96 \pm 0.381$ | $49.36 \pm 1.103$ | $25.92 \pm 1.173$ |
| Agnostic-SAM | $\mathbf{66.64 \pm 0.657}$ | $\mathbf{61.13 \pm 0.636}$ | $\mathbf{52.26 \pm 0.502}$ | $\mathbf{27.66 \pm 1.265}$ |
| ASAM | $66.88 \pm 0.593$ | $61.53 \pm 0.487$ | $52.77 \pm 0.561$ | $30.33 \pm 1.788$ |
| Agnostic-ASAM | $\mathbf{67.38 \pm 0.106}$ | $\mathbf{62.72 \pm 0.304}$ | $\mathbf{54.58 \pm 0.572}$ | $\mathbf{32.77 \pm 0.388}$ |
| **Dataset CIFAR-10** | | | | |
| SGD | $89.98 \pm 0.070$ | $84.83 \pm 0.085$ | $75.06 \pm 0.385$ | $54.47 \pm 1.265$ |
| SAM | $91.26 \pm 0.007$ | $88.19 \pm 1.060$ | $83.43 \pm 0.622$ | $61.69 \pm 0.289$ |
| FSAM | $91.35 \pm 0.318$ | $87.58 \pm 0.353$ | $82.78 \pm 2.057$ | $58.09 \pm 2.276$ |
| Agnostic-SAM | $\mathbf{92.38 \pm 0.007}$ | $\mathbf{90.20 \pm 0.318}$ | $\mathbf{85.33 \pm 0.268}$ | $\mathbf{70.02 \pm 0.403}$ |
| ASAM | $91.98 \pm 0.007$ | $89.24 \pm 0.572$ | $84.39 \pm 0.445$ | $64.82 \pm 6.880$ |
| Agnostic-ASAM | $\mathbf{92.06 \pm 0.367}$ | $\mathbf{90.01 \pm 0.282}$ | $\mathbf{86.09 \pm 0.657}$ | $\mathbf{73.25 \pm 0.353}$ |

All experiments are conducted using the ResNet32 architecture, with models trained from scratch for 200 epochs. The batch size is set to 512 for the training set and 128 for the validation set. Following Li et al. (2024) and Foret et al. (2021), we set $\rho = 0.1$ and $\rho_1 = 2\rho_2 = 0.2$ for SAM, FSAM, and Agnostic-SAM, $\rho_1 = 2\rho_2 = 2\rho = 2.0$ for ASAM and Agnostic-ASAM when training with all noise levels, except for the 80%, where we reduce the perturbation radius by half to ensure more stable convergence. In line with Li et al. (2024), we apply additional cutout techniques along with the basic augmentations outlined in Section 4.1. Each experiment is repeated three times with different random seeds, and we report the average and standard deviation of the results in Table 5. Note that training with SGD is prone to overfitting as the number of epochs increases. Therefore, we present the best results for SGD training at both 200 and 400 epochs.

### 4.4 EXPERIMENTS ON META-LEARNING

The concept of Agnostic-SAM is inspired by the agnostic approach in the MAML setting, where the meta-model is optimized on the meta-training set but aims to minimize loss on the validation set. The key difference is that Agnostic-SAM uses the gradient from the validation set as an indicator to close the generalization gap between the training and testing sets. Despite this difference, both approaches share the same underlying principle, making it reasonable to expect that applying Agnostic-SAM in the MAML setting will result in improved generalization performance.

Table 6: Meta-learning results on Mini-Imagenet dataset. All baseline results are taken from Abbas et al. (2022)

| Method | Accuracy | |
|---|---|---|
| | 5 ways 1 shot | 5 ways 5 shots |
| MAML | 47.13 | 62.20 |
| SHARP-MAML | 49.72 | 63.18 |
| Agnostic-SAM | **50.08** | **64.29** |

We compare our approach to standard MAML and Sharp-MAML (Abbas et al., 2022), which also addresses the loss-landscape flatness in bilevel models. The experiments follow the setup from

Table 7: Meta-learning results on Omniglot dataset. All baseline results are taken from Abbas et al. (2022)

| Method | Accuracy | |
|---|---|---|
| | 20 ways 1 shot | 20 ways 5 shots |
| MAML | 91.77 | 96.16 |
| SHARP-MAML | **92.89** | 96.59 |
| Agnostic-SAM | 92.66 | **97.28** |

Abbas et al. (2022), specifically using the Sharp-MAML$_{low}$ variation, which focuses on minimizing the sharpness of meta-models trained on the meta-training set. Note that during the testing phase of MAML, only the meta-training set is used for a few update steps of the meta-model; and our Agnostic-SAM approach incorporates both the training and validation sets in the meta-model training process. Ideally, both the meta-training and meta-validation sets should be utilized to minimize the lower-level loss during training. However, this could introduce inconsistencies between the training and testing phases, potentially degrading performance during testing. To avoid this issue, we duplicate the meta-training set and use it as a validation set to minimize the lower-level loss of the meta-model, applying this procedure consistently during both the training and testing phases.

As with other experiments, we set $\rho_1 = 2\rho_2 = 2\rho$, with $\rho$ as the perturbation radius for Sharp-MAML$_{low}$, and report the results in Tables 6 and 7. Our method consistently outperforms most baselines with significant improvements, demonstrating the effectiveness of Agnostic-SAM and its flexibility across various settings.

## 5 ABLATION STUDY

### 5.1 COSINE SIMILARITY OF GRADIENTS

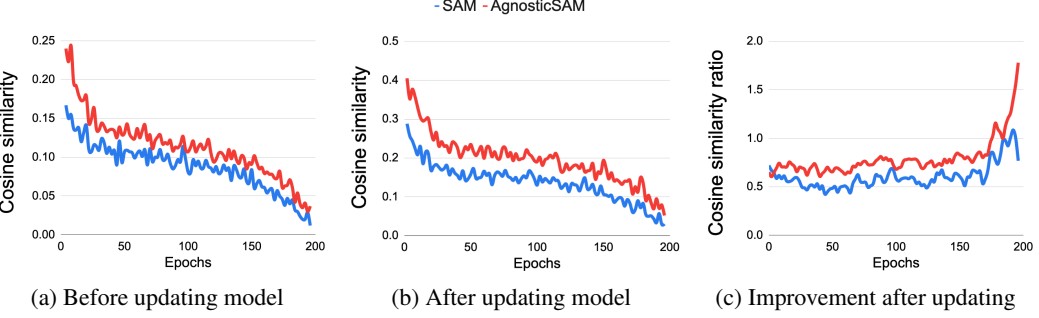

(a) Before updating model  (b) After updating model  (c) Improvement after updating

Figure 1: Cosine similarity of two gradients $\nabla_\theta \mathcal{L}_{B^t}(\theta_l)$ and $\nabla_\theta \mathcal{L}_{B^v}\left(\tilde{\theta}_l^v\right)$ **(a)** before updating model $cosine_b$, **(b)** after updating model $cosine_a$ and **(c)** the improvement of this score $change$

In Theorem 2, we prove that minimizing the loss function $\mathcal{L}_{B^t}$ could encourage two gradients $\nabla_\theta \mathcal{L}_{B^t}\left(\tilde{\theta}_l^t\right)$ and $\nabla_\theta \mathcal{L}_{B^v}\left(\tilde{\theta}_l^v\right)$ to be more congruent since our update aims to maximize its lower bound, which is $\nabla_\theta \mathcal{L}_{B^t}(\theta_l) \cdot \nabla_\theta \mathcal{L}_{B^v}\left(\tilde{\theta}_l^v\right)$. In this subsection, we measure the cosine similarity between two gradients $\nabla_\theta \mathcal{L}_{B^t}(\theta_l)$ and $\nabla_\theta \mathcal{L}_{B^v}\left(\tilde{\theta}_l^v\right)$ before (denoted as $cosine_b$) and after (denoted as $cosine_a$) updating the model and measure the change of these two score (denoted as $change$).

$$cosine_b = \frac{\nabla_\theta \mathcal{L}_{B^t}(\theta_l) \cdot \nabla_\theta \mathcal{L}_{B^v}\left(\tilde{\theta}_l^v\right)}{\|\nabla_\theta \mathcal{L}_{B^t}(\theta_l)\|_2 \|\nabla_\theta \mathcal{L}_{B^v}\left(\tilde{\theta}_l^v\right)\|_2}$$

$$cosine_a = \frac{\nabla_\theta \mathcal{L}_{B^t}\left(\theta_{l+1}\right) \cdot \nabla_\theta \mathcal{L}_{B^v}\left(\tilde{\theta}^v_{l+1}\right)}{\|\nabla_\theta \mathcal{L}_{B^t}\left(\theta_{l+1}\right)\|_2 \|\nabla_\theta \mathcal{L}_{B^v}\left(\tilde{\theta}^v_{l+1}\right)\|_2}$$

$$change = \frac{cosine_a - cosine_b}{cosine_a}$$

As shown in Figure 1c, both SAM and Agnostic-SAM improve the similarity after updating the model, this improvement also increases across training epochs. However, the similarity score of our Agnostic-SAM is always higher than SAM across the training process both before and after updating the model. This is evident that our Agnostic-SAM encourages gradient in training and validation set to be more similar during the training process.

## 5.2 Effectiveness of hyper-parameters

**Momentum factor $\beta$.** As mentioned in section 3.3, we use momentum with a factor $\beta$ to estimate the gradients of the validation set. This approach helps stabilize the training process and ensures the model minimizes the loss across the entire validation set, rather than just a mini-batch. In this subsection, we examine the effect of the momentum factor on the model's performance. When setting $\beta = 0$, the perturbed model in each iteration maximizes the loss on a mini-batch of the training set while minimizing the loss on a mini-batch of the validation set. When $\beta > 0$, the perturbed model aims to minimize the loss over the entire validation set, while maximizing the loss on a mini-batch of the training set.

The experiments are set up with the same hyper-parameters as those of experiments on CIFAR100 under noisy labels settings in Section 4.3 with basic data augmentation but without the cutout technique. We set $\rho = 0.1$ for SAM and $\rho_1 = 2\rho_2 = 0.2$ for Agnostic-SAM. Results in Table 8 show that the value of $\beta$ does not significantly affect model performance overall. As such, we simply set $\beta = 0.9$ in all experiments. With $\beta = 0$, our method still outperforms baselines consistently, strengthening our idea of using validation gradient to indicate the model into wider local minima while reducing the generalization gap of training and testing datasets.

Table 8: Effectiveness of momentum factor $\beta$ on performance

| Method | SAM | Agnostic-SAM | | | | |
|---|---|---|---|---|---|---|
| | | 0.0 | 0.3 | 0.5 | 0.7 | 0.9 |
| Accuracy | $70.31 \pm 0.2$ | $71.14 \pm 0.3$ | $71.12 \pm 0.1$ | $70.865 \pm 0.2$ | $70.76 \pm 0.3$ | $70.91 \pm 0.3$ |

**Validation batch size $|B^v|$ and complexity; sensitivity of perturbation radius $\rho_1$ and $\rho_2$.** Detail of these experiment is presented in Appendix A.2

## 6 Conclusion and Limitation

In this paper, we explore the relationship between Sharpness-Aware Minimization (SAM) and the underlying principles of the Model-Agnostic Meta-Learning (MAML) algorithm, specifically in terms of their effects on model generalization. Building on this connection, we integrate sharpness-aware minimization with the agnostic perspective from MAML to develop a novel optimization framework, introducing the Agnostic-SAM approach. This method optimizes the model toward wider local minima using training data while ensuring low loss values on validation data. As a result, Agnostic-SAM demonstrates enhanced robustness against data shift issues. Through extensive experiments, we empirically show that Agnostic-SAM consistently outperforms baseline methods, delivering significant improvements in model performance across various datasets and challenging tasks. One limitation to note is that using an additional validation set when finding the perturbed model could potentially increase training time (depending on the size of the validation set). We consider this a trade-off between performance and training complexity. However, this issue could potentially be mitigated by reusing gradients from the training set in previous steps and we leave this as a direction for future work to reduce training complexity and still maintain performance.

## Reproducibility Statement

We provide details of hyper-parameters for each experiment in Section 4 and Appendix A.2. Additionally, we open-source our code and provide instructions, scripts, and log files to reproduce experiments at `https://anonymous.4open.science/r/AgnosticSAM-F17F/README.md`

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

## A  APPENDIX / SUPPLEMENTAL MATERIAL

In this appendix, we present the proofs in our paper and additional experiments. We open-source our code and provide instruction, scripts, and log files to reproduce experiments at `https://anonymous.4open.science/r/AgnosticSAM-F17F/README.md`

### A.1  ALL PROOFS

**Proof of Theorem 1**

*Proof.* We use the PAC-Bayes theory in this proof. In PAC-Bayes theory, $\theta$ could follow a distribution, says $P$, thus we define the expected loss over $\theta$ distributed by $P$ as follows:

$$\mathcal{L}_{\mathcal{D}}(\theta, P) = \mathbb{E}_{\theta \sim P}\big[\mathcal{L}_{\mathcal{D}}(\theta)\big]$$
$$\mathcal{L}_{\mathcal{S}}(\theta, P) = \mathbb{E}_{\theta \sim P}\big[\mathcal{L}_{\mathcal{S}}(\theta)\big].$$

For any distribution $P = \mathcal{N}(\mathbf{0}, \sigma_P^2 \mathbb{I}_k)$ and $Q = \mathcal{N}(\theta, \sigma^2 \mathbb{I}_k)$ over $\theta \in \mathbb{R}^k$, where $P$ is the prior distribution and $Q$ is the posterior distribution, use the PAC-Bayes theorem in Alquier et al. (2016a), for all $\beta > 0$, with a probability at least $1 - \delta$, we have

$$\mathcal{L}_{\mathcal{D}}(\theta, Q) \leq \mathcal{L}_{\mathcal{S}}(\theta, Q) + \frac{1}{\beta}\Big[\mathsf{KL}(Q\|P) + \log \frac{1}{\delta} + \Psi(\beta, N)\Big], \tag{9}$$

where $\Psi$ is defined as

$$\Psi(\beta, N) = \log \mathbb{E}_P \mathbb{E}_{\mathcal{D}^N}\Big[ \exp\big\{ \beta\big[\mathcal{L}_{\mathcal{D}}(f_\theta) - \mathcal{L}_{\mathcal{S}}(f_\theta)\big]\big\}\Big].$$

When the loss function is bounded by $L$, then

$$\Psi(\beta, N) \leq \frac{\beta^2 L^2}{8N}.$$

The task is to minimize the second term of RHS of (9), we thus choose $\beta = \sqrt{8N}\frac{\mathsf{KL}(Q\|P)+\log\frac{1}{\delta}}{L}$. Then the second term of RHS of (9) is equal to

$$\sqrt{\frac{\mathsf{KL}(Q\|P) + \log\frac{1}{\delta}}{2N}} \times L.$$

The KL divergence between $Q$ and $P$, when they are Gaussian, is given by formula

$$\mathsf{KL}(Q\|P) = \frac{1}{2}\left[\frac{k\sigma^2 + \|\theta\|^2}{\sigma_P^2} - k + k\log\frac{\sigma_P^2}{\sigma^2}\right].$$

For given posterior distribution $Q$ with fixed $\sigma^2$, to minimize the KL term, the $\sigma_P^2$ should be equal to $\sigma^2 + \|\theta\|^2/k$. In this case, the KL term is no less than

$$k\log\Big(1 + \frac{\|\theta_0\|^2}{k\sigma^2}\Big).$$

Thus, the second term of RHS is

$$\sqrt{\frac{\mathsf{KL}(Q\|P) + \log\frac{1}{\delta}}{2N}} \times L \geq \sqrt{\frac{k\log\big(1 + \frac{\|\theta\|^2}{k\sigma^2}\big)}{4N}} \times L \geq L$$

when $\|\theta\|^2 > \sigma^2\big\{\exp(4N/k) - 1\big\}$. Hence, for any $\|\theta\|_2 > \sigma^2\big\{\exp(4N/k) - 1\big\}$, we have the RHS is greater than the LHS, and the inequality is trivial. In this work, we only consider the case:

$$\|\theta\|^2 < \sigma^2\big(\exp\{4N/k\} - 1\big). \tag{10}$$

Distribution $P$ is Gaussian centered around $\mathbf{0}$ with variance $\sigma_P^2 = \sigma^2 + \|\theta\|^2/k$, which is unknown at the time we set up the inequality, since $\theta$ is unknown. Meanwhile, we have to specify $P$ in advance,

since $P$ is the prior distribution. To deal with this problem, we could choose a family of $P$ such that its means cover the space of $\theta$ satisfying inequality (10). We set

$$c = \sigma^2\big(1 + \exp\{4N/k\}\big)$$

$$P_j = \mathcal{N}\big(0, c\exp\frac{1-j}{k}\mathbb{I}_k\big)$$

$$\mathfrak{P} := \big\{P_j : j = 1, 2, \dots\big\}$$

Then the following inequality holds for a particular distribution $P_j$ with probability $1 - \delta_j$ with $\delta_j = \frac{6\delta}{\pi^2 j^2}$

$$\mathbb{E}_{\theta' \sim \mathcal{N}(\theta, \sigma^2)}\mathcal{L}_\mathcal{D}\big(f_{\theta'}\big) \leq \mathbb{E}_{\theta' \sim \mathcal{N}(\theta, \sigma^2)}\mathcal{L}_\mathcal{S}\big(f_{\theta'}\big) + \frac{1}{\beta}\left[\mathsf{KL}(Q\|P_j) + \log\frac{1}{\delta_j} + \Psi(\beta, N)\right].$$

Use the well-known equation: $\sum_{j=1}^{\infty}\frac{1}{j^2} = \frac{\pi^2}{6}$, then with probability $1 - \delta$, the above inequality holds with every $j$. We pick

$$j^* := \left\lfloor 1 - k\log\frac{\sigma^2 + \|\theta\|^2/k}{c}\right\rfloor = \left\lfloor 1 - k\log\frac{\sigma^2 + \|\theta\|^2/k}{\sigma^2(1 + \exp\{4N/k\})}\right\rfloor.$$

Therefore,

$$1 - j^* = \left\lceil k\log\frac{\sigma^2 + \|\theta\|^2/k}{c}\right\rceil$$

$$\Rightarrow \quad \log\frac{\sigma^2 + \|\theta\|^2/k}{c} \leq \frac{1 - j^*}{k} \leq \log\frac{\sigma^2 + \|\theta_0\|^2/k}{c} + \frac{1}{k}$$

$$\Rightarrow \quad \sigma^2 + \|\theta\|^2/k \leq c\exp\left\{\frac{1 - j^*}{k}\right\} \leq \exp(1/k)\big[\sigma^2 + \|\theta\|^2/k\big]$$

$$\Rightarrow \quad \sigma^2 + \|\theta\|^2/k \leq \sigma_{P_{j^*}}^2 \leq \exp(1/k)\big[\sigma^2 + \|\theta\|^2/k\big].$$

Thus the KL term could be bounded as follow

$$\mathsf{KL}(Q\|P_{j^*}) = \frac{1}{2}\left[\frac{k\sigma^2 + \|\theta\|^2}{\sigma_{P_{j^*}}^2} - k + k\log\frac{\sigma_{P_{j^*}}^2}{\sigma^2}\right]$$

$$\leq \frac{1}{2}\left[\frac{k(\sigma^2 + \|\theta\|^2/k)}{\sigma^2 + \|\theta\|^2/k} - k + k\log\frac{\exp(1/k)\big(\sigma^2 + \|\theta\|^2/k\big)}{\sigma^2}\right]$$

$$= \frac{1}{2}\left[k\log\frac{\exp(1/k)\big(\sigma^2 + \|\theta\|^2/k\big)}{\sigma^2}\right]$$

$$= \frac{1}{2}\left[1 + k\log\left(1 + \frac{\|\theta_0\|^2}{k\sigma^2}\right)\right]$$

For the term $\log\frac{1}{\delta_{j^*}}$, with recall that $c = \sigma^2\big(1 + \exp(4N/k)\big)$ and

$j^* = \left\lfloor 1 - k\log\frac{\sigma^2 + \|\theta\|^2/k}{\sigma^2(1 + \exp\{4N/k\})}\right\rfloor$, we have

$$\log\frac{1}{\delta_{j^*}} = \log\frac{(j^*)^2\pi^2}{6\delta} = \log\frac{1}{\delta} + \log\left(\frac{\pi^2}{6}\right) + 2\log(j^*)$$

$$\leq \log\frac{1}{\delta} + \log\frac{\pi^2}{6} + 2\log\left(1 + k\log\frac{\sigma^2\big(1 + \exp(4N/k)\big)}{\sigma^2 + \|\theta\|^2/k}\right)$$

$$\leq \log\frac{1}{\delta} + \log\frac{\pi^2}{6} + 2\log\left(1 + k\log\big(1 + \exp(4N/k)\big)\right)$$

$$\leq \log\frac{1}{\delta} + \log\frac{\pi^2}{6} + 2\log\left(1 + k\big(1 + \frac{4N}{k}\big)\right)$$

$$\leq \log\frac{1}{\delta} + \log\frac{\pi^2}{6} + \log(1 + k + 4N).$$

Hence, the inequality

$$\mathcal{L}_{\mathcal{D}}\Big(\theta', \mathcal{N}(\theta, \sigma^2 \mathbb{I}_k)\Big) \leq \mathcal{L}_{\mathcal{S}}\Big(\theta', \mathcal{N}(\theta, \sigma^2 \mathbb{I}_k)\Big) + \sqrt{\frac{\mathsf{KL}(Q\|P_{j^*}) + \log \frac{1}{\delta_{j^*}}}{2N}} \times L$$

$$\leq \mathcal{L}_{\mathcal{S}}\Big(\theta', \mathcal{N}(\theta, \sigma^2 \mathbb{I}_k)\Big)$$

$$+ \frac{L}{2\sqrt{N}} \sqrt{1 + k \log \Big(1 + \frac{\|\theta\|^2}{k\sigma^2}\Big) + 2 \log \frac{\pi^2}{6\delta} + 4 \log(N+k)}$$

$$\leq \mathcal{L}_{\mathcal{S}}\Big(\theta', \mathcal{N}(\theta, \sigma^2 \mathbb{I}_k)\Big)$$

$$+ \frac{L}{2\sqrt{N}} \sqrt{k \log \Big(1 + \frac{\|\theta\|^2}{k\sigma^2}\Big) + O(1) + 2 \log \frac{1}{\delta} + 4 \log(N+k)}.$$

Since $\|\theta' - \theta\|^2$ is $k$ chi-square distribution, for any positive $t$, we have

$$\mathbb{P}\big(\|\theta' - \theta\|^2 - k\sigma^2 \geq 2\sigma^2 \sqrt{kt} + 2t\sigma^2)\big) \leq \exp(-t).$$

By choosing $t = \frac{1}{2} \log(N)$, with probability $1 - N^{-1/2}$, we have

$$\|\theta' - \theta\|^2 \leq \sigma^2 \log(N) + k\sigma^2 + \sigma^2 \sqrt{2k \log(N)} \leq k\sigma^2 \Big(1 + \sqrt{\frac{\log(N)}{k}}\Big)^2.$$

By setting $\sigma = \rho \times \big(\sqrt{k} + \sqrt{\log(N)}\big)^{-1}$, we have $\|\theta' - \theta\|^2 \leq \rho^2$. Hence, we get

$$\mathcal{L}_{\mathcal{S}}\Big(\theta', \mathcal{N}(\theta, \sigma^2 \mathbb{I}_k)\Big) = \mathbb{E}_{\theta \sim \mathcal{N}(\theta, \sigma^2 \mathbb{I}_k)} \mathbb{E}_{\mathcal{S}}\big[f_{\theta'}\big] = \int_{\|\theta'-\theta\| \leq \rho} \mathbb{E}_{\mathcal{S}}\big[f_{\theta'}\big] d\mathcal{N}(\theta, \sigma^2 \mathbb{I})$$

$$+ \int_{\|\theta'-\theta\| > \rho} \mathbb{E}_{\mathcal{S}}\big[f_{\theta'}\big] d\mathcal{N}(\theta, \sigma^2 \mathbb{I})$$

$$\leq \Big(1 - \frac{1}{\sqrt{N}}\Big) \max_{\|\theta'-\theta\| \leq \rho} \mathcal{L}_{\mathcal{S}}(\theta') + \frac{1}{\sqrt{N}} L$$

$$\leq \max_{\|\theta'-\theta\|_2 \leq \rho} \mathcal{L}_{\mathcal{S}}(\theta') + \frac{2L}{\sqrt{N}}.$$

It follows that

$$\mathcal{L}_{\mathcal{D}}(\theta) \leq \max_{\|\theta'-\theta\| \leq \rho} \mathcal{L}_{\mathcal{S}}(\theta') + \frac{4L}{\sqrt{N}} \Bigg[ \sqrt{k \log \Big(1 + \frac{\|\theta\|^2}{\rho^2} \big(1 + \sqrt{\log(N)/k}\big)^2\Big)}$$

$$+ 2\sqrt{\log \big(\frac{N+k}{\delta}\big) + O(1)} \Bigg]$$

$$= \mathcal{L}_{\mathcal{D}}(\theta \mid \mathcal{S}) + \frac{4L}{\sqrt{N}} \Bigg[ \sqrt{k \log \Big(1 + \frac{\|\theta\|^2}{\rho^2} \big(1 + \sqrt{\log(N)/k}\big)^2\Big)}$$

$$+ 2\sqrt{\log \big(\frac{N+k}{\delta}\big) + O(1)} \Bigg].$$

By choosing $\theta = \theta^*$ and $\mathcal{S} = S^v$ hence $N = N^v$, we reach the conclusion. $\qquad\square$

**Proof of Theorem 2**

*Proof.* We have

$$\mathcal{L}_{B^t}\Big(\tilde{\theta}_l^t\Big) = \mathcal{L}_{B_t}(\theta_l) + \eta_1 \|\nabla_\theta \mathcal{L}_{B^t}(\theta_l)\|_2^2 - \eta_2 \nabla_\theta \mathcal{L}_{B^t}(\theta_l) \cdot \nabla_\theta \mathcal{L}_{B^v}\Big(\tilde{\theta}_l^v\Big).$$

This follows that

$$\nabla_\theta \mathcal{L}_{B^t}\left(\tilde{\theta}_l^t\right) = \nabla_\theta \mathcal{L}_{B_t}\left(\theta_l\right) + 2\eta_1 H_{B^t}\left(\theta_l\right)\nabla_\theta \mathcal{L}_{B^t}\left(\theta_l\right)$$

$$- \eta_2\left[H_{B^t}\left(\theta_l\right)\nabla_\theta \mathcal{L}_{B^v}\left(\tilde{\theta}_l^v\right) + H_{B^v}\left(\tilde{\theta}_l^v\right)\nabla_\theta \mathcal{L}_{B^t}\left(\theta_l\right)\right],$$

where $H_{B^t}\left(\theta_l\right) = \nabla_\theta^2 \mathcal{L}_{B_t}\left(\theta_l\right)$ and $H_{B^v}\left(\tilde{\theta}_l^v\right) = \nabla_\theta^2 \mathcal{L}_{B^v}\left(\tilde{\theta}_l^v\right)$ are the Hessian matrices.

$$\nabla_\theta \mathcal{L}_{B^v}\left(\tilde{\theta}_l^v\right) \cdot \nabla_\theta \mathcal{L}_{B^t}\left(\tilde{\theta}_l^t\right) = \nabla_\theta \mathcal{L}_{B_t}\left(\theta_l\right) \cdot \nabla_\theta \mathcal{L}_{B^v}\left(\tilde{\theta}_l^v\right)$$

$$+ 2\eta_1 \nabla_\theta \mathcal{L}_{B^v}\left(\tilde{\theta}_l^v\right)^T H_{B^t}\left(\theta_l\right)\nabla_\theta \mathcal{L}_{B^t}\left(\theta_l\right)$$

$$- \eta_2 \nabla_\theta \mathcal{L}_{B^v}\left(\tilde{\theta}_l^v\right)^T H_{B^t}\left(\theta_l\right)\nabla_\theta \mathcal{L}_{B^v}\left(\tilde{\theta}_l^v\right)$$

$$- \eta_2 \nabla_\theta \mathcal{L}_{B^v}\left(\tilde{\theta}_l^v\right)^T H_{B^v}\left(\tilde{\theta}_l^v\right)\nabla_\theta \mathcal{L}_{B^t}\left(\theta_l\right).$$

We now choose $\eta_1 \leq \frac{\left|\nabla_\theta \mathcal{L}_{B_t}(\theta_l)\cdot\nabla_\theta \mathcal{L}_{B^v}(\tilde{\theta}_l^v)\right|}{12\left|\nabla_\theta \mathcal{L}_{B^v}(\tilde{\theta}_l^v)^T H_{B^t}(\theta_l)\nabla_\theta \mathcal{L}_{B^t}(\theta_l)\right|}$, we then have

$$\eta_1 \left|\nabla_\theta \mathcal{L}_{B^v}\left(\tilde{\theta}_l^v\right)^T H_{B^t}\left(\theta_l\right)\nabla_\theta \mathcal{L}_{B^t}\left(\theta_l\right)\right| \leq \frac{1}{12}\left|\nabla_\theta \mathcal{L}_{B_t}\left(\theta_l\right)\cdot\nabla_\theta \mathcal{L}_{B^v}\left(\tilde{\theta}_l^v\right)\right|.$$

This further implies

$$\eta_1 \nabla_\theta \mathcal{L}_{B^v}\left(\tilde{\theta}_l^v\right)^T H_{B^t}\left(\theta_l\right)\nabla_\theta \mathcal{L}_{B^t}\left(\theta_l\right) \geq -\frac{1}{12}\left|\nabla_\theta \mathcal{L}_{B_t}\left(\theta_l\right)\cdot\nabla_\theta \mathcal{L}_{B^v}\left(\tilde{\theta}_l^v\right)\right|.$$

Next we choose $\eta_2 \leq \min\left\{\frac{\left|\nabla_\theta \mathcal{L}_{B_t}(\theta_l)\cdot\nabla_\theta \mathcal{L}_{B^v}(\tilde{\theta}_l^v)\right|}{6\left|\nabla_\theta \mathcal{L}_{B^v}(\tilde{\theta}_l^v)^T H_{B^t}(\theta_l)\nabla_\theta \mathcal{L}_{B^v}(\tilde{\theta}_l^v)\right|}, \frac{\left|\nabla_\theta \mathcal{L}_{B_t}(\theta_l)\cdot\nabla_\theta \mathcal{L}_{B^v}(\tilde{\theta}_l^v)\right|}{6\left|\nabla_\theta \mathcal{L}_{B^v}(\tilde{\theta}_l^v)^T H_{B^v}(\tilde{\theta}_l^v)\nabla_\theta \mathcal{L}_{B^t}(\theta_l)\right|}\right\},$ we then have

$$\eta_2 \left|\nabla_\theta \mathcal{L}_{B^v}\left(\tilde{\theta}_l^v\right)^T H_{B^t}\left(\theta_l\right)\nabla_\theta \mathcal{L}_{B^v}\left(\tilde{\theta}_l^v\right)\right| \leq \frac{\left|\nabla_\theta \mathcal{L}_{B_t}\left(\theta_l\right)\cdot\nabla_\theta \mathcal{L}_{B^v}\left(\tilde{\theta}_l^v\right)\right|}{6}.$$

$$-\eta_2 \nabla_\theta \mathcal{L}_{B^v}\left(\tilde{\theta}_l^v\right)^T H_{B^t}\left(\theta_l\right)\nabla_\theta \mathcal{L}_{B^v}\left(\tilde{\theta}_l^v\right) \geq -\frac{\left|\nabla_\theta \mathcal{L}_{B_t}\left(\theta_l\right)\cdot\nabla_\theta \mathcal{L}_{B^v}\left(\tilde{\theta}_l^v\right)\right|}{6}.$$

$$\eta_2 \left|\nabla_\theta \mathcal{L}_{B^v}\left(\tilde{\theta}_l^v\right)^T H_{B^v}\left(\tilde{\theta}_l^v\right)\nabla_\theta \mathcal{L}_{B^t}\left(\theta_l\right)\right| \leq \frac{\left|\nabla_\theta \mathcal{L}_{B_t}\left(\theta_l\right)\cdot\nabla_\theta \mathcal{L}_{B^v}\left(\tilde{\theta}_l^v\right)\right|}{6}.$$

$$-\eta_2 \nabla_\theta \mathcal{L}_{B^v}\left(\tilde{\theta}_l^v\right)^T H_{B^v}\left(\tilde{\theta}_l^v\right)\nabla_\theta \mathcal{L}_{B^t}\left(\theta_l\right) \geq -\frac{\left|\nabla_\theta \mathcal{L}_{B_t}\left(\theta_l\right)\cdot\nabla_\theta \mathcal{L}_{B^v}\left(\tilde{\theta}_l^v\right)\right|}{6}.$$

Finally, we yield

$$\nabla_\theta \mathcal{L}_{B^v}\left(\tilde{\theta}_l^v\right) \cdot \nabla_\theta \mathcal{L}_{B^t}\left(\tilde{\theta}_l^t\right) \geq \nabla_\theta \mathcal{L}_{B_t}\left(\theta_l\right) \cdot \nabla_\theta \mathcal{L}_{B^v}\left(\tilde{\theta}_l^v\right) - \frac{1}{2}\left|\nabla_\theta \mathcal{L}_{B_t}\left(\theta_l\right)\cdot\nabla_\theta \mathcal{L}_{B^v}\left(\tilde{\theta}_l^v\right)\right|.$$

$\square$

## A.2    ADDITIONAL EXPERIMENTS

**Validation batch size $|B^v|$ and complexity**    Our method is to use a gradient on the validation set as a helper indicator to lead the model to wider local minima while maintaining low loss on the validation set, and the model should be updated mainly using training samples. Increasing validation mini-batch size could potentially increase performance and training time. In Table 9, we present the results of Agnostic-SAM with various validation batch sizes $|B^v|$ of CIFAR-100 with Resnet32 while maintaining a fixed training batch size $|B^t| = 512$, the other hyper-parameters are the same as above experiments with momentum factor $\beta$. We consider performance and training complexity to be the trade-off of Agnostic-SAM and find that setting $|B^t| = 4|B^v|$ works well for all experiments.

Table 9: Experiments on different sizes of validation mini-batch with a fixed size of training mini-batch is 512 samples

| Method | Validation batch-size | Accuracy | Training time (s/epochs) |
|---|---|---|---|
| SAM | 0 | $70.31 \pm 0.233$ | 11s |
| Agnostic-SAM | 16 | $70.58 \pm 0.219$ | 11s |
| | 32 | $71.07 \pm 0.212$ | 12s |
| | 64 | $70.67 \pm 0.049$ | 13s |
| | 128 | $71.21 \pm 0.056$ | 14s |
| | 256 | $71.04 \pm 0.219$ | 15s |

**Sensitivity of perturbation radius $\rho_1$ and $\rho_2$**   Throughout this paper, we used a consistent setting of $\rho_1 = 2\rho_2 = 2\rho$, where $\rho$ represents the perturbation radius in the SAM method for all experiments. While these hyperparameters could be optimized for each experiment individually, we find that this configuration delivers good performance across most experiments. By setting $\rho_1 > \rho_2$, we ensure that the perturbed model prioritizes maximizing the loss on the training set rather than minimizing it on the validation set. This approach encourages the model to focus primarily on minimizing sharpness during the actual update step in Formula 6.

To verify the impact of these hyperparameters on model performance, we conduct experiments with varying perturbation radius and present the results in Figure 2. Notably, the configuration where $\rho_1 > \rho_2$ consistently yields higher accuracy compared to the setting where $\rho_1 < \rho_2$. When increasing $\rho_2$, the model places more emphasis on minimizing the validation set loss, rather than sharpness on the training set during the actual update step in Formula 6. This shift in focus can lead to overfitting, ultimately reducing performance.

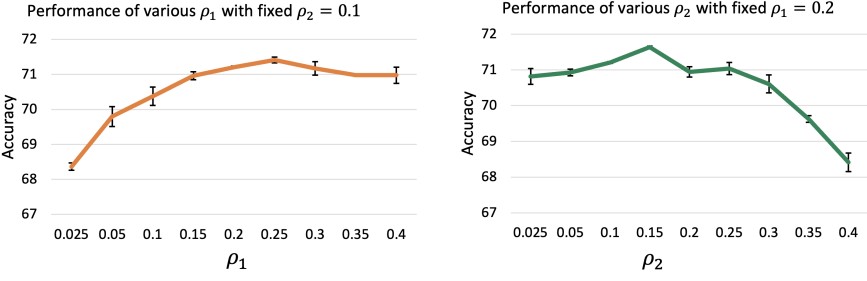

Figure 2: Experiments of various perturbation radius $\rho_1$ and $\rho_2$

**Analysis of loss landscape and eigenvalues of the Hessian matrix**   We demonstrate the effectiveness of Agnostic-SAM in guiding models toward flatter regions of the loss landscape, as compared to both SAM and SGD, in Figures 3 and 4. The loss landscapes are visualized with the same setting, the blue areas represent lower loss values, while the red areas indicate higher loss values. Although SAM is shown to lead the model to a flatter region than SGD, Agnostic-SAM achieves an even smoother and significantly flatter loss landscape, especially in experiments with EfficientNet-B2 in Figure 3.

To further validate that Agnostic-SAM successfully locates minima with low curvature, we compute the Hessian of the loss landscape and report the five largest eigenvalues, sorted from $\lambda_1$ to $\lambda_5$, in Table 10. These eigenvalues provide insight into the curvature of the model at the optimized parameters. Larger eigenvalues indicate steeper curvature, meaning the model is more sensitive to small changes in its parameters. Conversely, smaller eigenvalues suggest flatter minima, which are typically associated with improved robustness, better generalization, and reduced sensitivity to overfitting. Negative eigenvalues indicate non-convex curvature in certain directions.

As shown in Table 10, Agnostic-SAM consistently achieves positive and lower eigenvalues compared to the baseline methods, suggesting that it effectively leads the model toward flatter regions of the loss

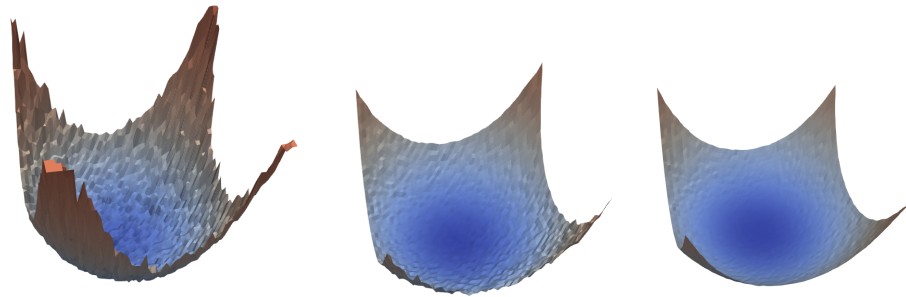

Figure 3: Loss landscape of **EffecientNet-B2** trained on Flower102 dataset with **(left)** SGD, **(middle)** SAM, and **(right)** Agnostic-SAM.

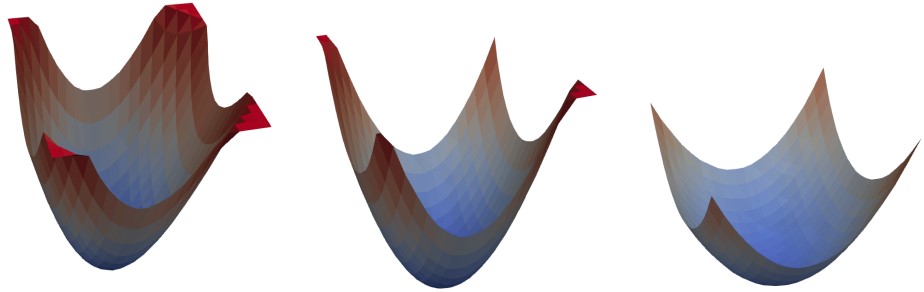

Figure 4: Loss landscape of **ResNet32** trained **(left)** SGD, **(middle)** SAM, and **(right)** Agnostic-SAM on Cifar100 dataset.

| Methods | Top-5 eigenvalues of Hessian matrix | | | | |
| --- | --- | --- | --- | --- | --- |
| | $\lambda_1$ | $\lambda_2$ | $\lambda_3$ | $\lambda_4$ | $\lambda_5$ |
| **EfficientNet-B2 on Flower102** | | | | | |
| SGD | $2.05 \times 10^5$ | $0.45 \times 10^5$ | $0.26 \times 10^5$ | $-0.47 \times 10^5$ | $-0.49 \times 10^5$ |
| SAM | $1.61 \times 10^3$ | $1.34 \times 10^3$ | $1.23 \times 10^3$ | $1.04 \times 10^3$ | $-0.97 \times 10^3$ |
| Agnostic-SAM | $0.61 \times 10^3$ | $0.41 \times 10^3$ | $0.37 \times 10^3$ | $0.32 \times 10^3$ | $0.31 \times 10^3$ |
| **Resnet32 on Cifar100** | | | | | |
| SGD | $3.07 \times 10^6$ | $2.40 \times 10^6$ | $2.10 \times 10^6$ | $1.64 \times 10^6$ | $1.46 \times 10^6$ |
| SAM | $1.50 \times 10^6$ | $1.14 \times 10^6$ | $0.96 \times 10^6$ | $0.87 \times 10^6$ | $0.81 \times 10^6$ |
| Agnostic-SAM | $1.04 \times 10^6$ | $0.79 \times 10^6$ | $0.66 \times 10^6$ | $0.58 \times 10^6$ | $0.57 \times 10^6$ |

Table 10: Eigenvalues of Hessian matrix

landscape. These results further support the efficacy of Agnostic-SAM in optimizing for smoother and more stable solutions across a variety of architectures and tasks.

