# OpenReview forum: "Agnostic Sharpness-Aware Minimization"
_ICLR.cc/2025/Conference — ICLR 2025 Conference Withdrawn Submission_

### Official Review · Reviewer_XCb4 · 2024-10-30

**Soundness:** 2
**Presentation:** 2
**Contribution:** 1
**Rating:** 3
**Confidence:** 5

**Summary:**

The authors propose Agnostic-SAM, a new variant of the Sharpness-Aware Minimization (SAM) algorithm. Instead of only the batchwise gradient ascent perturbation step of SAM, Agnostic-SAM additionally performs a descent step on a validation batch, before computing the gradient for the final update. The authors motivate their work from a PAC-Bayes bound and report experimental results on image classification tasks (vanilla classification, noisy labels, meta-learning).

**Strengths:**

- The proposed method requires an additional hyperparameter, but the authors found a way of setting it consistently throughout their experiments: $ \rho_{1} = 2 \rho_{2} $.

- Agnostic-SAM improves over baselines in most cases (even though I have doubts about the setups, see below)

- Combining ideas from MAML and SAM is a creative approach

**Weaknesses:**

**Comparison to Baselines**

At its core, Agnostic-SAM changes the perturbation step of SAM by adding an additional perturbation based on gradients from a separate, smaller data batch, and the authors claim improved generalization performance. However, several methods have proposed adjustments to SAM’s perturbation model with improved generalization performance. Most similar to Agnostic-SAM, [1] adds random perturbations to the gradient-based perturbation, while [2] and [3] perform multi-step perturbations. Many more methods exist ([6,7,8,...]), but none of those appear as baselines in the experiments. The only other standard baselines are in Table 2 (ASAM), and in Table 5 (ASAM and FSAM). In the MAML experiment (Tables 6 and 7) the authors report improved performance over [9]. However, they only show the Sharp-MAML-low version from [9], even though other versions exist. In particular, the Sharp-MAML-both variant from [9] outperforms Agnostic-SAM in three out of the four reported cases. It is unclear why this is not reported or explained.


**Training time**

The proposed method requires two additional forward-backward passes on the validation batch, which leads to increased computational cost compared to SAM (roughly 27\% wall clock time according to Table 9). While the authors briefly mention this in the conclusion, a more thorough discussion and evaluation is needed, as this affects the fairness of comparisons in the main paper.


**Hyperparameters and train settings**

The authors report some baseline values from the original papers, and others are reproduced with the $\rho$ values from those papers. For instance, for WRN28-10 on CIFAR100 the SGD number is taken from the SAM paper [5], and the SAM number is reproduced with the same $\rho$ value, but is lower than the number from the SAM paper (83.0\% vs 83.5\% in the SAM paper, which would outperform the reported Agnostic-SAM number). Similar observations hold for ASAM, and CIFAR100. Lower reproduced numbers can be due to different training settings and are not a problem per se if the comparison is fair, but here I have certain doubts because the optimal $\rho$ value can be sensitive to the training settings and was taken from the reference papers. Further, some choices, like e.g. $\rho=0.05$ for SAM in ImageNet transfer learning while $\rho=0.1$ for ImageNet training from scratch look just arbitrary. Some $\rho$ tuning must have taken place, since the authors even claim that _accuracies tend to decrease when reducing $\rho$_. Further, it is unclear how exactly the authors came up with the choice $\rho_{1}=2\rho_{2}$ and if it is based on the ablation in A2, purely by intuition or additional experiments and tuning. Finally, the scope of the experiments is somewhat limited. In particular, there are no experiments with VisionTransfomers, no experiments on text data, and the only larger-scale experiments (ImageNet) are with fairly weak models (at most ResNet-34 for training from scratch).


**Theorem 1**

The authors present Theorem 1 as a central motivation for their method. However, this theorem is nearly identical to Theorem 1 and its proof in the original SAM paper [5], with minimal modification. As with [5], this theorem would theoretically motivate a version of SAM based on average-case rather than worst-case perturbations. The presented generalization bound implies an average-case sharpness bound, which is only subsequently upper-bounded by a worst-case sharpness bound. This limitation was already present in [5] and has since been highlighted, for example, in [4]. Furthermore, the conclusions from this theorem, i.e. why exactly it would motivate equation (3) and the final algorithm, are not understandable to me.


**Clarity**

Apart from the disconnect between Theorem 1 and the method, it is not well justified why exactly the alignment of the gradients of the perturbed points from train and validation batches would be beneficial for generalization, especially since in the experiments both batches are from the train set. Overall, the MAML perspective is unclear to me, since in the practical algorithm, train and validation batches are both sampled from the train set, and there is only one task to solve in almost all experiments. Additional confusion arises from unclear terminology  (e.g. the notation $\theta^{*}(\theta)$ wasn’t introduced, the Taylor expansion in (7) is presented as an exact equality, etc.)



[1] Yong Liu, Siqi Mai, Minhao Cheng, Xiangning Chen, Cho-Jui Hsieh, & Yang You (2022). Random Sharpness-Aware Minimization. In Advances in Neural Information Processing Systems.

[2] Kim, H., Park, J., Choi, Y., Lee, W., and Lee, J. Exploring the effect of multi-step ascent in sharpness-aware minimization

[3] Goncalo Mordido, Pranshu Malviya, Aristide Baratin, & Sarath Chandar (2024). Lookbehind-SAM: k steps back, 1 step forward. In Forty-first International Conference on Machine Learning.
[4] Maksym Andriushchenko and Nicolas Flammarion (2022). Towards Understanding Sharpness-Aware Minimization. ICML 2022

[5] Pierre Foret, Ariel Kleiner, Hossein Mobahi, and Behnam Neyshabur. Sharpness-aware minimization for efficiently improving generalization. In ICLR, 2021

[6] Minyoung Kim, Da Li, Shell X Hu, and Timothy Hospedales. Fisher SAM: Information geometry and sharpness aware minimisation. In Kamalika Chaudhuri, Stefanie Jegelka, Le Song, Csaba Szepesvari, Gang Niu, and Sivan Sabato (eds.), Proceedings of the 39th International Conference on Machine Learning

[7] Mi, P.; Shen, L.; Ren, T.; Zhou, Y.; Sun, X.; Ji, R.; and Tao,D. 2022. Make Sharpness-Aware Minimization Stronger: A Sparsified Perturbation Approach

[8] Jiawei Du, Hanshu Yan, Jiashi Feng, Joey Tianyi Zhou, Liangli Zhen, Rick Siow Mong Goh, & Vincent Tan (2022). Efficient Sharpness-aware Minimization for Improved Training of Neural Networks. In International Conference on Learning Representations.

[9] Momin Abbas, Quan Xiao, Lisha Chen, Pin-Yu Chen, and Tianyi Chen. Sharp-maml: Sharpness-aware model-agnostic meta learning

**Questions:**

According to Section 3.3 the goal is to align the perturbed gradients of the validation and train batches. Why do the authors then report the alignment of the unperturbed gradient of the train batch with the perturbed gradient of the validation batch in Section 5.1?

---

### Official Review · Reviewer_UCdS · 2024-11-01

**Soundness:** 2
**Presentation:** 2
**Contribution:** 2
**Rating:** 3
**Confidence:** 5

**Summary:**

The paper introduces Agnostic-SAM, an optimization method that integrates insights from MAML into SAM. The approach seeks to update the model to a region that not only minimizes sharpness on the training set but also implicitly ensures strong performance on the validation set.

**Strengths:**

The paper provides a comprehensive evaluation of Agnostic-SAM across a wide range of tasks, including image classification, transfer learning, training with label noise, and meta-learning.

**Weaknesses:**

1.	The motivation for the problem formulation in Equation 3 is not convincingly justified. It would benefit from a clearer explanation of why this specific formulation was chosen and how it directly leads to generalization.
2.	The paper does not sufficiently clarify how the integration of MAML’s insights with the proposed problem formulation and algorithm specifically aids generalization. A deeper theoretical or empirical justification is needed.
3.	The proposed algorithm assumes the existence of a held-out validation set. However, in practice, the training set is used as the validation set, which diverges from the theoretical framework. This discrepancy is particularly problematic in datasets like CIFAR-10 and CIFAR-100, where the training loss converges to zero, a behavior not typically observed with a true validation set.
4.	From Algorithm 1, it appears that Agnostic-SAM requires double the computational time compared to SAM. In the experiments, SAM is compared by allowing SGD to run for double the iterations for fair comparison [1]. It would be fairer to allow SAM and ASAM to run for twice the iterations of Agnostic-SAM in the experiment.

[1] Foret, P., Kleiner, A., Mobahi, H., and Neyshabur, B. Sharpness-aware minimization for efficiently improving generalization. In International Conference on Learning Representations, 2021

**Questions:**

Please refer to the concerns raised in the weaknesses section above.

---

### Official Review · Reviewer_cgjs · 2024-11-02

**Soundness:** 2
**Presentation:** 2
**Contribution:** 1
**Rating:** 3
**Confidence:** 4

**Summary:**

This paper combines MAML into SAM, proposing a new optimization scheme to improve generalization performance. The paper provides a theoretical propositions on generalization bound and gradients alignments, but it is regarded that the paper mainly focuses on verifying its generalization effectiveness numerically measured on some deep learning tasks. The paper also provides additional ablation results to support gradient alignments and momentum.

**Strengths:**

- SAM and MAML are both found to be effective for enhancing generalization performance, and that the paper is attempting to explore the intersection of these is encouraging.
- The paper follows a standard procedure to evaluate the proposed method (Agnostic-SAM) and shows its effectiveness in experiments.

**Weaknesses:**

There are several concerns on this paper summarized as follows.

Method
- The main idea and motivation of this work, as its current form, remain quite random. They are two of many potential ways to improve generalization performance, but without clearly justifying why these two, the paper simply combine the two approaches and end up providing experimental results. This diminishes the technical contributions and novelty.
- The authors also claim that it is a "framework", but with it being the simple combination of SAM and MAML, it has not been rigorously evaluated to be a framework as to whether this can serve as a general scheme so it remains as an initial idea. There have been many advancements since the original SAM and MAML, but the paper only takes a proof-of-concept approach, limiting its potential.
- This idea requires additional computations (validation set, additional forward-backward, hyperparameter tuning) but it is unclear whether this is worth, in particular, compared to other potential ways to improve generalization.

Experiments
- The experiments are also a bit bland without being tailored to specifically analyze any aspect of SAM and MAML simply evaluating the final performances, lacking novelty and interesting insights.
- The proposed scheme is only compared to naive baselines, and it is seen that the improvements are very marginal across many experiments. It is a bit critical in the sense that Agnostic-SAM makes use of the additional validation set and more computations to get validation gradients, which leaves a question that whether Agnostic-SAM is really the best possible choice for generalization.

**Questions:**

- Can authors state exactly what contributions to claim from Theorems 1 and 2?
- (on the first Imagenet experiments) the top-1/top-5 accuracies seem quite low, why is that the case? can the authors also provide Resnet50 results? how many runs are these results? can authors provide standard errors?
- It is unclear the exact difference between two versions of Agnostic-SAM (Table 2 and Table 5); if it means using different base SAM (i.e., SAM or ASAM), it appears that Agnostic-SAM (with ASAM) often underperforms Agnostic-SAM (with SAM), why is it the case?
- How did author come up with the rules to set perturbation bounds originally?

---

### Official Review · Reviewer_hqiX · 2024-11-09

**Soundness:** 2
**Presentation:** 2
**Contribution:** 2
**Rating:** 3
**Confidence:** 3

**Summary:**

This work aims to combine Sharpness-Aware Minimization with Model-Agnostic Meta-Learning, by having worst-case robustified versions of the loss in both the inner and outer loop of meta-learning. This is then tested in the usual supervised learning setups in vision and some meta-learning benchmarks, where the method is shown to outperform the baselines.

**Strengths:**

- The motivation to combine the element of sharpness-minimization in meta-learning for better generalization makes sense. This is the operationalized well in the form of an algorithm that is shown to perform slightly better with the baselines.

- The method seems to be extensively tested in supervised learning setups, meta-learning scenarios, as well as those with label noise.

**Weaknesses:**

- Difference wrt Abbas et al. 2022: When compared with this prior work, it is unclear what is the novelty here. The authors mention this paper, but don't bother to explain the similarities or differences. The method here looks **eerily similar** to the two-year old prior work, which is arguably better written and presented and a lot more richer. Except for little bits of analysis on congruence between gradients, I can't spot much of a methodological difference.

- Supervised learning experiments: In departure to their motivation, the authors start presenting results on supervised learning. I understand that this can be simulated in the meta-learning setup as well, but it comes across as confusing. Then their method involves 4 gradient computations per step, while SGD and SAM will involve 1 and 2 gradient computations respectively. So for a fairer comparison, the authors should have reported results with letting the baselines have more compute. Thus, given the excessive runtime, the method does not seem worth the effort of obtaining marginal gains.

- Ablation studies on the relevance for SAM in inner/outer stages of meta-learning would have been insightful: Which part benefits more from SAM? Can the authors run an ablation study?

- Momentum hyperparameter and it's ablation: The Table 8 would suggest that having no momentum results in better performance, but it is bizarre that the authors continue to keep using a momentum in all their results, despite of that. Especially when the improvements they report are not infrequently of the similar range.

**Questions:**

besides the above, I am curious why the perturbation radii are set the way they are, ie. inner rho twice the outer rho? Was it grid -searched? is there some intuition behind this setting?

---

### Note · Authors · 2024-11-18

I have read and agree with the venue's withdrawal policy on behalf of myself and my co-authors.